# Seminal Plasma Transcriptome and Proteome: Towards a Molecular Approach in the Diagnosis of Idiopathic Male Infertility

**DOI:** 10.3390/ijms21197308

**Published:** 2020-10-03

**Authors:** Rossella Cannarella, Federica Barbagallo, Andrea Crafa, Sandro La Vignera, Rosita A. Condorelli, Aldo E. Calogero

**Affiliations:** Department of Clinical and Experimental Medicine, University of Catania, 95123 Catania, Italy; federica.barbagallo11@gmail.com (F.B.); crafa.andrea@outlook.it (A.C.); sandrolavignera@unict.it (S.L.V.); rosita.condorelli@unict.it (R.A.C.)

**Keywords:** seminal fluid transcriptome, seminal fluid proteome, male idiopathic infertility, oligozoospermia

## Abstract

As the “-omic” technology has largely developed, its application in the field of medical science seems a highly promising tool to clarify the etiology, at least in part, of the so-called idiopathic male infertility. The seminal plasma (SP) is made-up of secretions coming from the male accessory glands, namely epididymis, seminal vesicles, and prostate. It is not only a medium for sperm transport since it is able to modulate the female reproductive environment and immunity, to allow the acquisition of sperm competence, to influence the sperm RNA content, and even embryo development. The aim of this systematic review was to provide an updated and comprehensive description of the main transcripts and proteins reported by transcriptome and proteome studies performed in the human SP of patients with idiopathic infertility, in the attempt of identifying possible candidate molecular targets. We recurrently found that micro RNA (miR)-34, miR-122, and miR-509 are down-regulated in patients with non-obstructive azoospermia (NOA) and oligozoospermia compared with fertile controls. These molecules may represent interesting targets whose predictive role in testicular sperm extraction (TESE) or assisted reproductive techniques (ART) outcome deserves further investigation. Furthermore, according to the available proteomic studies, ECM1, TEX101, lectingalactoside-binding andsoluble 3 binding protein (LGALS3BP) have been reported as accurate predictors of TESE outcome. Interestingly, ECM1 is differently expressed in patients with different ART outcomes. Further prospective, ample-sized studies are needed to validate these molecular targets that will help in the counseling of patients with NOA or undergoing ART.

## 1. Introduction

Infertility has been defined by the World Health Organization (WHO) as the inability to achieve conception after 12–24 months of sexual unprotected intercourse (WHO, 1983). Its prevalence has increased through the decades and currently affects 10% to 15% of couples worldwide [1]. The awareness of the male factor in the pathogenesis of couple infertility has increased. Accordingly, the male partner has been esteemed to be involved, alone or in combination, in the 50% of infertility cases and, therefore, the concomitant investigation of both partners is wise and recommended in the management of couple infertility.

Sperm analysis is the first-level exam to be requested to the male partner. It provides information on sperm number, motility, morphology, presence of somatic cells, and macroscopic features such as volume, pH, and viscosity, which are indicative of the health and integrity of male accessory glands and seminal ducts [2]. However, the sperm analysis does not give enough information to recognize the cause of male infertility. Indeed, spermatozoa may be incompetent to fertilize the oocyte despite normal conventional sperm parameters in a significant number of patients [3]. This has encouraged scientists to search for the so-called “bio-functional” sperm parameters, such as sperm mitochondrial membrane potential, late or early apoptosis, degree of chromatin compactness, and sperm DNA fragmentation [4], which represent second level sperm parameters providing information on sperm function. Among them, sperm DNA fragmentation is the most well-accepted parameter, and the available data confirm its impact on the pregnancy outcome [5,6,7]. However, these parameters lack validation, and they are not all widely accepted and used, and, still now, the cause of male factor infertility remains unexplained in a not negligible percentage of cases [8].

As the “-omic” technology has broadly developed, its application in the field of medical science has been claimed as a highly promising tool to clarify the cause of at least a part of apparently idiopathic male infertility. The seminal plasma (SP) is made-up of secretions arising from the male accessory glands, namely epididymis, seminal vesicles, and prostate. The SP is not only a medium for sperm transport, since it can modulate the female reproductive environment and immunity [9] to allow the acquisition of sperm competence [10] to influence sperm RNA content, and even embryo development, as transcriptome studies reveal [11]. In recent years, thousands of specific transcripts and proteins have been found in human seminal plasma (SP). This systematic review aims to provide an updated and comprehensive description of the main transcripts and proteins reported by transcriptome and proteome studies performed in the human SP, in the attempt of identifying possible candidate molecular targets of male infertility.

## 2. Methods

### 2.1. Sources

This study was performed extracting data through extensive research in Pubmed, MEDLINE, Cochrane, Academic One Files, Google Scholar, and Scopus databases from their inception to August 2020. 

The search strategy included the following combination of Medical Subjects Headings (MeSH) terms and keywords: “seminal fluid proteome”, “seminal plasma proteome”, “seminal fluid transcriptome”, “seminal plasma transcriptome”, “male idiopathic infertility”, “oligozoospermia”, “azoospermia”, “fertilization”, “embryo development”, “pregnancy”. Other articles were extracted from the reference lists of the articles found by entering the aforementioned keywords.

### 2.2. Study Selection 

The study included all the articles that evaluated differences in expression of SP transcriptome or proteome in infertile patients and fertile controls. Moreover, all the studies correlating SP transcriptome or proteome with oocyte fertilization, embryo development, and assisted reproductive technique (ART) outcome were also included. 

Reviews, comments, letters to the editors, and animal studies on SP transcriptomics or proteomics were not included in this review.

## 3. Results

The articles collected using the aforementioned search strategy were discussed considering first those dealing with SP transcriptome and the difference in RNA expression in SP between infertile patients and fertile controls. A proposal for target RNAs was made, when possible. A similar framework was used for SP proteome.

### 3.1. Seminal Plasma Transcriptome

Transcriptome is defined as the overall content of RNA of a definite specimen, including coding RNAs—such as messenger RNAs (mRNAs), and non-coding RNAs—such as micro-RNAs (miRNAs), small interfering RNAs (siRNAs), antisense RNAs,piwi-interacting RNAs (piRNAs), and long non-coding RNAs (lncRNAs), which are engaged in the modulation of gene expression. Furthermore, the diction of small RNA (sRNA) has been used to refer to miRNAs, transfer RNA-derived fragments (tRFs), and piRNAs. 

Spermatozoa have been considered as a vehicle of the paternal packaged genome for decades. However, thousands of RNAs have been found in its cytoplasm [12]. In particular, current knowledge reports the presence of ~270 RNAs in human spermatids and human mature spermatozoa [13]. Although their function is still unclear, they are supposed to play a role in embryo formation and development [14]. Accordingly, the removal of sperm RNA by pre-treatment with an RNA-ase makes spermatozoa incapable of fertilization in mice [15]. 

About ~400 RNAs come from the human testis. The human SP, which is believed to provide proteins needed for sperm capacitation and to keep spermatozoa in a quiescent state, contains ~700 extracellular RNAs and, among them, ~400 originate from the seminal vesicles and prostate [13]. According to Gene Ontology enrichment analysis, these RNAs are involved in the regulation of vesicle-mediated transport, in protein kinases inhibition, and cellular response to zinc [13]. 

Interestingly, SPRNAs have been found to influence the content of sperm RNA. Particularly, epididymosomes are extracellular vesicles released by the epididymal epithelium, which encapsulate a macromolecular complex made up of proteins (required to avoid RNA degradation) and RNAs. During the sperm transit into the epididymis, epididymosomes interact with spermatozoa by modulating their RNA profile [11]. In more detail, while miRNAs are the most represented class of sRNA in the caput epididymis, tRFs are more abundantly found in the cauda [16,17]. During the epididymal transit, sperm miRNAs account for 50% of the global sRNA in the caput epididymis to the 16% in the cauda epididymis. By contrast, sperm tRF account for 24% to 65% in the caput and in the cauda epididymis, respectively [16,17]. Similarly, sRNA content of epididymosomes varies from the caput to the cauda region (Figure 1). This confirms that the content of sperm RNAs is modulated during the epididymal transit: spermatozoa are enriched of miRNAs in the caput, whereas they are enriched of tRFs in the cauda. This is notable, since the RNA placed in the caput epididymis (which is a component of the SP) is required for oocyte fertilization. Accordingly, a study in mice showed that spermatozoa retrieved from the caput epididymis is unable to penetrate the oocyte. It becomes competent for fertilization only by the addiction of RNAs of the cauda [16]. This evidence suggests a role of SP transcriptome in human fertility. The results coming from human transcriptome studies in fertile men and infertile patients are discussed below.

#### 3.1.1. Implication in Male Infertility

Ever since the existence of transcriptome in SPwas discovered, some experts have tried to understand its role in the pathogenesis of male infertility to understand the physio-pathological aspects and to identify possible diagnostic targets. 

A study carried out in the highest sample size investigated so far suggested that a differential expression of miRNAs in SP occurs between fertile men and infertile patients. The authors enrolled 1378 patients with idiopathic infertility and 486 fertile controls. The *miR-196a-2* CC genotype was found significantly associated with idiopathic male infertility, and CC patients had higher levels of miR-196a-5p in their SP. In vitro experiments have shown that this miRNA enhances germ cell apoptosis, which was prevented by its inhibition, in-vitro. Hence, the authors interestingly concluded that the miR-196a-2 CC polymorphism enhances SP miR-196a-5p levels, which, in turn, predispose to male idiopathic infertility by favoring germ cell apoptosis [18].

A miRNA expressed in the human SP has recently been proven to regulate germ cell apoptosis. Particularly, a study carried out in 30 patients with idiopathic infertility and high levels of sperm DNA fragmentation index (DFI) and 30 fertile controls found a significant down-regulation of the miR-424 in the seminal plasma of patients compared with controls. Subsequently, the authors analyzed the effects of the inhibition of miR-322 (that is the murine orthologue of the human miR-424) in a murine germ cell culture. The inhibition resulted in an up-regulation of pro-apoptotic genes and in the down-regulation of anti-apoptotic pathways, thus leading to germ cell apoptosis. The *Ddx3x* gene was reported as the direct target of the miR-322, which down-regulates this gene. This is of importance since the *Ddx3x* gene regulates embryo development. In particular, its upregulation leads to accumulation of p53 and embryo apoptosis. Interestingly, Che and collaborators reported that *Ddx3x* gene and its protein are expressed in human SP, but they resulted upregulated in the infertile group, where the miR-424 was down-regulated. Collectively, this evidence suggests that the down-regulation of the miR-424 in the SP of infertile patients could enhance germ cell apoptosis, leading to high sperm DFI. Moreover, by targeting the *Ddx3x* gene, this miRNA may be involved in embryo lethality [19]. Further evidence is needed to confirm whether it could be a reliable and accurate diagnostic target.

MiR-371a-3p has been described in human SP and it resulted directly correlated with sperm concentration and total sperm count [20]. This is in line with a great amount of evidence suggesting this miRNA as a high accurate (>90%) diagnostic target in testicular germ cell tumor [21]. Hence, it is supposable a role of the miR-371a-3p in the enhancement of germ cell proliferation, which could explain the direct correlation with the sperm number.

Another study has investigated whether a differential expression of piRNAs occurs in the SP of 211 infertile patients with asthenozoospermia or azoospermia and 91 fertile controls. Five piRNAs were found differently expressed with a high diagnostic potential at ROC curve [22]. They were piR-31068, piR-31925, piR-43771, piR-43773, and piR-31098. Their levels gradually decreased from fertile controls, to asthenozoospermic patients, and even further in patients with azoospermia. Due to the role of several piRNAs in germ cell development, the authors speculated that these piRNAs may be released in SP after the apoptosis of male germ cells (mainly, pachytene spermatocytes, and round spermatids) [22]. However, no study has so far validated the biological function of these piRNAs in spermatogenesis and, hence, the authors’ hypothesis still remains to be proved. 

In 96 patients with idiopathic male infertility (48 with oligozoospermia and 48 with non-obstructive azoospermia (NOA)), miR-19b and let-7a were found significantly up-expressed compared to the levels measured in 48 fertile controls [23]. Mir-19b has been shown to promote germ cell proliferation in-vitro [23]. Let-7 belongs to a family including nine mammalian miRNAs (let-7a to let-7i) with conserved sequences. Let-7 is highly expressed in the testis, where represents the 80% of the overall amount of miRNA in the juvenile testis, and the 11% of the adult one. Let-7a expression is elevated in germ cells, and it may inhibit proliferation [23]. To further confirm the role of let-7 as a possible diagnostic target in patients with infertility, a recent study reported the down-regulation of the hsa-let-7b-5p in spermatozoa of patients with asthenozoospermia compared to healthy controls [24]. In vitro analysis of germ cells demonstrated that this miRNA is capable of up-regulating the *Aurkb* gene, which is involved in glycolytic activities. Therefore, asthenozoospermia may result from the inhibition of glycolysis in patients with hsa-let-7b-5p down-regulation [24]. This confirm that let-7 family may represent a promising target in idiopathic male infertility.

Azoospermia deserves a careful management due to its psychological impact on the couple. The research has tried to find reliable markers to predict the recovery of spermatozoa from the testis. In this regard, the follicle-stimulating hormone (FSH) or testis histology have been suggested to have a prognostic value [25]. The study by Barcelò and colleagues was conducted to understand whether exosome miRNAs in human SP could address the origin of azoospermia and could help in predicting sperm retrieval [26]. The SP from 14 patients with NOA due to spermatogenic failure, 13 with obstructive azoospermia, 3 patients with severe oligozoospermia (<5 × 10^6^ mil/mL), and 9 fertile controls were analyzed. The authors reported a differential expression of 397 miRNAs among the analyzed samples. They then focused on the differences between NOA and obstructive azoospermia and reported the miR-31-5p as a predictive target of the origin of azoospermia, with an accuracy >90%. The sensitivity and specificity were even higher when FSH were added in the prediction model. Furthermore, in a logistic model the authors were able to demonstrate the effectiveness of miR-539-5p and miR-941 to predict the presence or not of spermatozoa within the testis [26]. 

Another study carried out in 100 patients with NOA and 100 fertile controls reported significantly higher levels of miR-141, miR-429, and miR-7-1-3p in the SP of patients compared to controls [27]. Their functional analysis showed that these miRNAs targeted genes triggering the apoptotic pathway in germ cells. Hence, in the investigated patients, NOA may be explained by an exaggerated pro-apoptotic mechanism occurring in spermatogenic cells [27].

A case-control study carried out in patients with NOA and varicocele assessed the levels of miR-192a in SP(the testicular tissue was also evaluated) in patients with sperm recovery (n = 27) and those were spermatozoa were not found in the semen after surgery (n = 33) (Zhi et al., 2018). Patients with sperm recovery showed lower levels of miR-192a compared to those without. miR-192a targets genes are involved in testicular hypoxia, venous hypertension, and oxidative stress, which are mechanisms usually induced by varicocele. Specifically, incubation of germ cells with the miR-192a is capable of increasing the Caspase-3 protein levels, which in turn enhances cell apoptosis [28]. The low levels of miR-192a may avoid germ cell apoptosis, thus explaining the sperm recovery after varicocelectomy in patients with NOA where this miRNA is down-regulated. Therefore, miR-192a has been suggested as a predictor of sperm recovery after varicocele repair in patients with NOA [28]. 

Similarly, a predictive value in testicular sperm retrieval has been assigned to the hsa-circ-0000116 [29].

This knowledge is useful since if further comprehensive studies on a greater number of patients will show the role of SP miRNAs as predictive markers of sperm retrieval, testicular biopsy (which is an invasive diagnostic technique currently requested to predict the outcome of testicular sperm extraction (TESE)) may be avoided in the future. 

Lastly, the role of the SP transcriptome as a possible target of infertility may not be restricted only to the impact on germ cell proliferation and sperm number, or to the prediction of sperm recovery in NOA, but also to the competence of sperm in fertilization. miRNAs carried in the SP can play an immunoregulatory role in the female reproductive tract [30]. In more detail, miR-223, miR-146a, miR-155, miR-23b, miR-17-92, and miR-34a have been proposed as possibly involved in the modulation of macrophages, Treg, and dendritic cell activity of the female genital tract to induce an immunotolerance and allowing sperm penetration in the female tract and embryo implantation in the endometrium. A dysregulation of these miRNAs has been observed in miscarriage, pre-eclampsia, and in small for gestational age (SGA) fetuses [30].

#### 3.1.2. Conclusion

The results of the aforementioned studies are summarized in Table 1. The majority of the studies carried out so far focused on miRNAs regulating germ cell proliferation or apoptosis. Human SP miRNA may be used in patients with idiopathic male infertility to predict testicular sperm retrieval (e.g., in NOA patients), or as markers of germ cell apoptosis in patients with spermatogenic failure. Particularly, three miRNAs are reported by more than two studies: miRNAs belonging to the miR-34, miR-122, and miR-509 families (Table 1). They all are down-regulated in patients with NOA and in those with oligozoospermia compared with fertile controls [26,31,32]. Hence, multi-center studies on ample cohorts should be accomplished to validate their diagnostic value in patients with idiopathic male infertility. Since the available knowledge is scanty at moment, it would be useful in the next future to develop proper studies focusing on the validation of a panel of SP RNAs predicting the TESE outcome.

### 3.2. Seminal Plasma Proteome

In the last decade, the field of proteomics has rapidly and significantly developed. The term “proteomics” was introduced in 1995 to define the large-scale identification of the protein components of a cell type, tissue, or biological fluid [34]. The birth of proteomics was inspired by the concept that a complete information cannot be obtained from the study of single genes and that the final product of a gene is more complex than the gene itself. The proteins are responsible for the phenotypes of the cells.

The great progress in proteomic technology has allowed us to identify protein biomarkers of diseases. Interestingly, proteomic analysis of SP may represent a precious tool for the evaluation of male infertility [35]. SP has a key role in the survival and the function of spermatozoa [36]. It has a very heterogeneous composition and is made by several proteins secreted by different glands [37]. The total volume of the ejaculate derives mostly from the seminal vesicles (65%) and prostate (25%), and for a minor part from testes and epididymis (10%) [38]. However, despite testes and epididymis contribute marginally for the total volume of the ejaculate, proteins with important functions in the SP are of testicular or epididymal origin [38]. Interestingly, in addition to soluble proteins, the SP comprehends also proteins contained in microvesicles, such as those released by epididymis (epididymosomes) and prostate (prostastomes), which seem to have a pivotal role in male fertility [37].

Proteomic technologies have rapidly advanced, progressing from basic electrophoresis techniques to liquid chromatography and mass spectrometry platform (LC-MS/MS). One of the first studies that analyzed a large number of proteins in the SP was published in 2006 [39]. They performed 2D electrophoresis followed by LC-MS/MS and found 923 proteins. A couple of years later, Batruch and colleagues identified more than 2000 proteins in the SP using offline multidimensional liquid chromatography and high-resolution mass spectrometry [38]. In 2012, Milardi and colleagues analyzed the SP of five fertile men by the modern LTQ-Orbitrap mass spectrometer to describe which proteins are common in fertile men [35]. 

Functionally, SP proteins are involved in several essential steps for spermatozoa survival and for fertilization [40]. The vast majority of the proteins present in the SP has been assigned a catalytic activity. In fact, most of these proteins such as the prostatic-specific antigen (PSA), are involved in the regulation, processing, or degradation of SP proteins and the coagulation of semen [35]. Among proteins with structural function, some are secreted by the seminal vesicles, the so-called gel-forming proteins: fibronectin (FN1), semenogelin-1 (SEMG1), and semenogelin-2 (SEMG2) [35]. SEMG1 is one of the most important proteins of human semen coagulum and it inhibits sperm progressive motility at ejaculation, until it is hydrolyzed by PSA [41]. Other studies reported that some SP proteins have a binding function [38], such as SP heparin-binding proteins (HBPs), which are involved in capacitation and in the interaction between gametes [42]. Albumin (ALB) and lactotransferrin (LTF) are also abundant in the SP and have transport functions. LTF is also known for its antimicrobial and antioxidant properties [35].

#### 3.2.1. Implication in Male Infertility

In the attempt to identify diagnostic targets of male infertility, some studies have focused on the proteomic pattern in human SP. Wang and colleagues have analyzed the SP samples of 38 patients with asthenozoospermia and 20 fertile controls, reporting a threefold down-regulation of 45 proteins and a threefold up-regulation of 56 proteins, in patients compared to controls [43]. These SP proteins are mainly produced by the epididymis and the prostate. Particularly, the authors addressed to the low levels of the DJ-1 protein (involved in the control of oxidative stress) a leading role in the pathogenesis of asthenozoospermia and infertility. Accordingly, high levels of reactive oxygen species (ROS) were found in patients, and the reduced levels of DJ-1 in the SP of asthenozoospermic patients could led to the incapacity of spermatozoa to face the oxidative damage [43]. 

Another study has assessed six plasma proteins (PSA, glucose, pepsinogen C, insulin-like growth factor binding protein-3, prostaglandin D synthase (PGDS), and BRCA1-like immunoreactive protein (BRCA1-LIP) in 202 SP samples from patients with azoospermia, oligozoospermia, vasectomy, and men with normozoospermia, by immunofluorimetry. The levels of PGDS were the only to correlate with sperm concentration, motility, and normal morphology. Moreover, its levels decreased gradually and significantly from normozoospermic, to oligozoospermic, azoospermic, and vasectomized patients. As PGDS is secreted into the SP by Sertoli cells, it has been suggested by the authors as a marker of obstruction [44]. However, it may also represent a marker of Sertoli cell dysfunction in patients with spermatogenic failure. Subsequently, other authors have addressed to the seminal PGDS a role in the improvement of sperm motility, thus claiming that this protein may impact on male fertility potential [45], although further multi-center studies are needed to confirm this.

Similarly, in a comparison study between normozoospermic, asthenozoospermic, oligozoospermic, and azoospermic patients, eight proteins (fibronectin, prostatic acid phosphatase (PAP), proteasome subunit alpha type-3, beta-2-microglobulin, galectin-3-binding protein, prolactin-inducible protein, and cytosolic nonspecific dipeptidase) were differently expressed. Specifically, higher levels (particularly of PAP) were found in azoospermic patients, whereas no difference was observed in the others. Hence, the authors suggested this panel of eight proteins as a marker of azoospermia [46].

Altogether, these studies indicate that a differential pattern of expression of SP proteins occurs in patients with quantitative or qualitative abnormalities of sperm parameters. Worryingly, a wide range of proteins are reported, often from a limited number of patients, thus restricting the applicability of the findings in the clinical practice. 

Of great utility are those studies aimed at finding markers which may differentiate the etiology of azoospermia (obstructive vs. non-obstructive) or that may predict the testicular histology (Sertoli cell-only-syndrome, maturation arrest, hypospermatogenesis). SP samples from 119 patients with normozoospermia or azoospermia were used to test the utility of 18 biomarkers for the differential diagnosis of spermatogenic failure. Interestingly, the authors reported the epididymal-expressed ECM1 and testis-expressed TEX101 as the most accurate markers to differentiate obstructive vs. non-obstructive forms. Particularly, the cut-off of 2.3 μg/mL of the EMC1 was shown to distinguish normal spermatogenesis from obstructive azoospermia, with a 100% specificity, and obstructive azoospermia from NOA with a 100% sensitivity and a 73% specificity. Furthermore, TEX101, at a threshold >5 ng/mL, could differentiate NOA underlined by Sertoli-cell only syndrome from NOA due to other testis histology (e.g., hypospermatogenesis, with a 67% specificity and a 100% sensitivity, or maturation arrest, with a 54% sensitivity and a 100% specificity) [47]. 

Another retrospective study searched for seminal predictors of residual spermatogenesis in 40 patients with NOA undergoing TESE. Significantly higher levels of lectingalactoside-binding, soluble 3 binding protein (LGALS3BP) in the SP were found in patients with a positive outcome of TESE. The cut-off of 153 ng/mL was reported with a sensitivity of 45% and a specificity of 100% [48].

Conversely, in a cohort of 40 patients with NOA, seminal anti-Müllerian hormone (AMH) resulted measurable in 23/40 patients (57.5% with a positive TESE outcome), and undetectable in 17/40 patients (58.2% with a positive TESE outcome). Hence, AMH levels have been indicated as not useful in predicting TESE outcome [49]. 

This line of evidence addresses to the SP proteome a possible predictive role of sperm recovery in NOA patients willing to undergo TESE. The clinical application of this would be highly relevant and, particularly, ECM1, TEX101, and LGALS3BP may represent interesting targets. Unfortunately, the number of studies on this are very limited and there is still a need for further studies to assess the consistency of this hypothesis. 

Notably, some effort has been made to evaluate whether the SP proteome may be useful in the prediction of the outcome of ART. The SP proteome from normozoospermic men belonging to the rescue intracytoplasmic sperm injection pregnancy group and the in-vitro fertilization pregnancy group was compared in the attempts to find determinants of ART outcome. Seventy-three proteins were found differently expressed (45 up-expressed and 28 down-expressed) between the two groups. Interestingly, ECM1 was among those differently expressed. Furthermore, lactoferrin (LTF), fibronectin (FN1), creatine kinase B type (CKB), kallikrein-2 (KLK2), aminopeptidase N (ANPEP), glycodelin (PAEP), alpha-1-antitrypsin (SERPINA1), and semenogelin-1 (SEMG1) were selected for validation as potentially interesting diagnostic future targets (Liu et al., 2020). This was the first study that focused on this very important aspect or reproductive medicine. Previous attempts have been done to evaluate the role of sperm proteome on ART [24,50] but no evaluation of SP was performed. 

#### 3.2.2. Conclusion

Several lines of evidence addressed to the SP proteome a role in male infertility. The practical need would be to find few accurate targets, to be used to predict TESE outcome in patients with NOA or ART successfulness. This is a very interesting input and, limited to the available information, ECM1, TEX101, and LGALS3BP may be selected for prospective ample-sized studies.

## 4. The Future of Seminal Plasma Biomarkers in Male Infertility 

Idiopathic male infertility has been esteemed to pertain to the ~70% of cases of oligozoospermia [8]. Basic science applied to clinical research (currently named translational medicine) may represent a promising strategy to understand the pathogenesis of several forms of apparently idiopathic male infertility [51,52]. 

The evidence outlined in this review highlights the possible role of specific SP transcriptomic or proteomic molecular targets in male idiopathic infertility.

In specific cases, SP RNAs could predict the appearance of spermatozoa in the ejaculate of patients with NOA, as it has been shown for miR-192a [28] and hsa-circ-0000116 [29]. Hence, RNAs regulating germ cell apoptosis and involved in spermatogenesis may also have a role for the prediction of sperm retrieval but further studies are needed. Indeed, no predictive analysis has been performed so far. Furthermore, no evidence on the possible predictive role of SP RNAs on ART outcome has been released. Adequately sized studies should be performed to cover this blackhole. In this view, the predictive potential of miR-34, miR-122, and miR-509 deserves to be investigated. 

More evidence is available for SP proteome, as the accuracy of ECM1, TEX101, and LGALS3BP in predicting TESE outcome in patients with NOA has already been analyzed. Very interestingly, ECM1 may play a role also in the prediction of ART outcome. Focused studies are warranted to confirm this evidence. Taken together, the findings on the studies we suggest need to be performed, will help in the counseling of patients with abnormal sperm parameters.

## Figures and Tables

**Figure 1 ijms-21-07308-f001:**
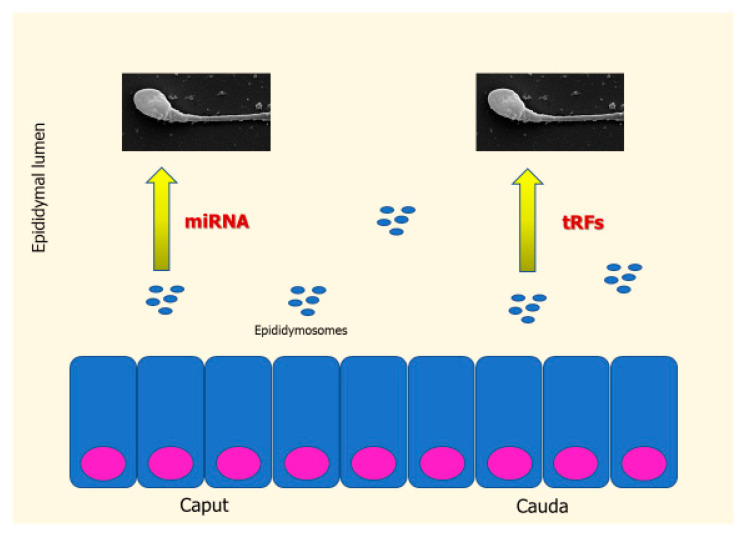
RNA exchange between epididymosomes and sperm in the epididymal lumen. In the caput epididymis, spermatozoa are enriched of miRNAs; in the cauda, they are enriched of RNA-derived fragments (tRFs). Rectangles indicate cells of epididymal epithelium. Blue circles indicate epididymosomes.

**Table 1 ijms-21-07308-t001:** Expression profile of human transcripts in seminal plasma of infertile patients and fertile controls.

Authors	Population	Results
[31]	118 patients with NOA and 168 fertile controls	Down-regulated miRNAs: hsa-miR-34c; hsa-miR-122; hsa-miR-509-5p
[23]	96 idiopathic infertile patients (48 with oligozoospermia and 48 with NOA) and 48 fertile controls	Up-regulated miRNAs: miR-19b and let-7a
[27]	100 patients with NOA and 100 fertile controls	Up-regulated miRNAs: miR-141, miR-429 and miR-7-1-3p
[22]	211 infertile patients (AT or azoospermia) and 91 fertile controls	Down-regulated piRNAs: piR-31068, piR-31925, piR-43771, piR-43773 and piR-31098
[18]	1378 patients with idiopathic infertility and 486 fertile controls	Down-regulated miRNA: miR-196a-5p
[26]	14 patients with NOA, 13 with OA and 9 normozoospermic controls	Down-regulated miRNAs (NOA): hsa-miR-202-3p; hsa-miR-514a-3p; hsa-miR-202-5p; hsa-miR-510-3-5p; hsa-miR-509-3-5p; hsa-miR-510-5p; hsa-miR-513c-5p; hsa-miR-518e-3p; hsa-miR-508-5p; hsa-miR-520h; hsa-miR-9-3p; hsa-miR-506-3p; hsa-miR-383-5p; hsa-miR-34c-5p; hsa-miR-517c-3p; hsa-miR-873-5p; hsa-miR-34b-5p; hsa-miR-513a-3p; hsa-miR-52; hsa-miR-452-5p; hsa-miR-122-5p; hsa-miR-449a; hsa-miR-449a-5p; hsa-miR-455-5p; has-miR-9-5p; 132-5p; hsa hsa-miR-203a Up-regulated miRNAs (NOA): hsa-miR-363-5p; hsa-miR-365a-3p; hsa-miR-550a-5p; 423-5p; hsa-miR-153-3p; hsa-miR-196b-3p; hsa-miR-96-5p; hsa-let-7-1-3p Down-regulated miRNAs (OA): hsa-miR-202-3p; hsa-miR-514a-3p; hsa-miR-202-5p; hsa-miR-510-3-5p; hsa-miR-509-3-5p; hsa-miR-510-5p; hsa-miR-513c-5p; hsa-miR-518e-3p; hsa-miR-508-5p; hsa-miR-520h; hsa-miR-9-3p; hsa-miR-506-3p; hsa-miR-383-5p; hsa-miR-34c-5p; hsa-miR-517-3p; hsa-miR-873-5p; hsa -miR-34b-5p; hsa -miR-513-3p; hsa -miR-52; hsa -miR-452-5p; hsa -miR-122-5p; hsa -miR-449a; hsa -miR-449a-5p; hsa -miR-455-5p; hsa -miR-819b; hsa -miR-890; hsa -miR-34c-3p; hsa -miR-891a-5p; hsa -miR-888-5p; hsa -miR-124-3p; hsa -miR-892a; hsa -miR-551b-3p; hsa -miR-424-5p; hsa -miR-181b-5p; hsa -miR-31-3p; hsa -miR-181a-5p; hsa -miR-31-5p; hsa -miR-10b-3p; hsa -miR-222-3p; hsa -miR-455-3p; hsa -miR-205-5p; hsa -miR-182-3p; hsa-miR-95-3p Up-regulated miRNAs (OA): hsa -miR-363-3p; hsa -miR-365a-3p; hsa -miR-29a-3p; hsa-miR-296-5p; hsa-miR-23b-5p; hsa-miR-21-3p; hsa-miR-193a-3p; hsa-miR-29c-3p; hsa-miR-361-3p
[19]	30 infertile patients with high sperm DFI and 30 fertile controls	Up-regulated miRNA: miR-424
[32]	40 patients with KS, 60 with NOA, 60 OA and 40 normozoospermic controls	Down-regulated miRNAs: has-miR-509-5p; has-miR-122-5p; has-miR-34b-3p; has-miR-34c-5p
[33]	17 patients with AT, 15 patients with teratozoospermia, 17 patients with AT, 18 normozoospermic infertile patients	Up-regulated miRNA: miRNA-582-5p (teratozoospermia and AT)

Abbreviations: AT, asthenoteratozoospermia; KS, Klinefelter syndrome; NOA, non-obstructive azoospermia; OA, oligoasthenoteratozoospermia.

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
