# Peer review of "Seminal Plasma Transcriptome and Proteome: Towards a Molecular Approach in the Diagnosis of Idiopathic Male Infertility"

_ijms, 2020, doi:10.3390/ijms21197308_

Round 1

Reviewer 1 Report

One of my concerns is that the authors mix up the data from the studies of sperm and SP proteins. While some proteins potentially might be engulfed by sperm and/or affect the intracellular events in sperm, most of the SP proteins support fertilization. For example, regulation of transcription has been mentioned several times, but it is unlikely that any transcription takes place after mature sperm are mixed with the SP proteins. In a similar manner, several sub-sections are devoted to trans-generation epigenetic effect of micro-RNAs, but such data were not demonstrated for the SP RNAs (only from sperm). The author should not list intracellular functions for SP proteins unless the effect within the sperm has been shown for specific proteins. In the same manner, were alternation of the proteins by oxidative stress (an addition off carbonyl groups, for example) demonstrated for the SP proteins? Some articles that are mentioned looked at both sperm and SP omics, and it is hard to understand from the text whether the findings were in sperm or SP.

The author mentions that SP proteins/RNAs can be potentially used for the screening of samples as a standardized method. Therefore, rather than just listing data from each condition as a separate table, it would make sense to compare the conditions. Don’t we expect to find some common markers in infertile patients, smokers, and patients with OS? If we don’t, then we need more studies and better protocol for optimization and standardization.

A paragraph on page 2 is repeated twice.

The author should mention what new they added to this review as compared to the review from their group in 2018.

Author Response

Answers to the Reviewer #1 comments

Manuscript ID ijms-937469 Revised

Comment 1: One of my concerns is that the authors mix up the data from the studies of sperm and SP proteins. While some proteins potentially might be engulfed by sperm and/or affect the intracellular events in sperm, most of the SP proteins support fertilization. For example, regulation of transcription has been mentioned several times, but it is unlikely that any transcription takes place after mature sperm are mixed with the SP proteins. In a similar manner, several sub-sections are devoted to trans-generation epigenetic effect of micro-RNAs, but such data were not demonstrated for the SP RNAs (only from sperm). The author should not list intracellular functions for SP proteins unless the effect within the sperm has been shown for specific proteins. In the same manner, were alternation of the proteins by oxidative stress (an addition off carbonyl groups, for example) demonstrated for the SP proteins? Some articles that are mentioned looked at both sperm and SP omics, and it is hard to understand from the text whether the findings were in sperm or SP.

Answer to comment 1: We appreciated your suggestion. Accordingly, the structure of the review was changed and only studies on seminal plasma transcriptome and proteome are now cited and discussed. An additional search was made to fully accomplish the purpose of this review article.

Comment 2: The author mentions that SP proteins/RNAs can be potentially used for the screening of samples as a standardized method. Therefore, rather than just listing data from each condition as a separate table, it would make sense to compare the conditions. Don’t we expect to find some common markers in infertile patients, smokers, and patients with OS? If we don’t, then we need more studies and better protocol for optimization and standardization.

Answer to comment 2: We did it. The targets more frequently reported (e.g. miR-34, miR-122, miR-509) or having an already shown predictive role (e.g. ECM1, TEX101, and LGALS3BP) were listed in the attempt to suggest a panel to be validated by further studies.

Comment 3: A paragraph on page 2 is repeated twice.

Answer to comment 3: Corrected.

Comment 4: The author should mention what new they added to this review as compared to the review from their group in 2018.

Answer to comment 4: We could not find any review on this topic published in 2018. If this Reviewer refers to Giacone et al. (World J Mens Health. 2019;37(2):148-156), this article has a different aim since it focused on sperm epigenetics (e.g. sperm DNA methylation and protamination).    

Reviewer 2 Report

In this Manuscript the authors analyze the compiled information on papers related to seminal plasma transcriptome and proteome in order to try and determine possible molecular determinants of idiopathic male infertility in humans.

This is an interesting and comprehensive effort that yields a lot of systematized information for an interested researcher. There are a few issues that the authors should consider in a revised version:

1- The tables should be uniform, only Table 2 includes the numbers of patients in each study, for example. Although that information may be in the text, this should be addressed.

2- In that regard: method validation and sample size should have been exclusion criteria, and the discussion of the data reviewed should be more critical. The authors do not discuss method validity and this can be relevant (as some papers are not very recent), and there are studies with very few patients that cannot be compared with larger cohorts, and I question if even should be included in a general analysis on this topic. The authors treat all the data as having potentially the same relevance, and this is not what a critical review should do. I would suggest eliminating (or moving to supplemental tables) articles that may have poor methodology or have very few samples so as to concentrate on the best data available. For example (p.4-5) Obesity involves 10 patients, smoking 5, endurance training 6. Are these numbers relevant? I don't think so, and I'm sure the authors don't either. The paper should therefore be carefully revised with this in mind.

Author Response

Answers to the Reviewer #2 comments

Manuscript ID ijms-937469 Revised

Comment 1: The tables should be uniform, only Table 2 includes the numbers of patients in each study, for example. Although that information may be in the text, this should be addressed.

Answer to comment 1: Thank you. The revised manuscript has now only one table, which reports the number and the characteristics of the patients enrolled in the various studies.

Comment 2: In that regard: method validation and sample size should have been exclusion criteria, and the discussion of the data reviewed should be more critical. The authors do not discuss method validity and this can be relevant (as some papers are not very recent), and there are studies with very few patients that cannot be compared with larger cohorts, and I question if even should be included in a general analysis on this topic. The authors treat all the data as having potentially the same relevance, and this is not what a critical review should do. I would suggest eliminating (or moving to supplemental tables) articles that may have poor methodology or have very few samples so as to concentrate on the best data available. For example (p.4-5) Obesity involves 10 patients, smoking 5, endurance training 6. Are these numbers relevant? I don't think so, and I'm sure the authors don't either. The paper should therefore be carefully revised with this in mind. 

Answer to comment 2: We appreciated this comment. In the revised manuscript, only articles on seminal plasma transcriptome and proteome were left and discussed. An additional search was made to accomplish the purpose of the present article. Articles with very few patients mainly focused with the topic of trans-generation inheritance, which, in line with the suggestions of another reviewer, was not included in the revised version of this manuscript. Also, the studies were compared and a practical proposal of some molecular candidates was made (please see paragraphs 3.1.2, 3.2.2 and 4).

Reviewer 3 Report

The structuring of the review is not appropriate.

The Interaction lacks an explanation why the authors have chosen keywords and terms used in "Methods" for the idiopathic male fertility. In addition, the count of publications do not correspond to reported numbers and differ from the diagram (Fig. 1). I do not consider it necessary to analyze and present such information at all in review.

The tables show only the list of genes and proteins without context relationship to pathologies. The authors should describe everything in the text, evaluate and discuss. The individual subchapters contain very little information related to the topic and a small set of works is often cited.

Discussion is not absolutely sufficient and very short. Additionally, obtained information from cited papers are not properly concluded.

Author Response

Answers to the Reviewer #3 comments

Manuscript ID ijms-937469 Revised

Comment 1: The structuring of the review is not appropriate.

Answer to comment 1: The structure of the review was entirely revised and only those studies on seminal plasma transcriptome and proteome were cited. An additional search was made to further accomplish the purpose of this review article.

Comment 2: The interaction lacks an explanation why the authors have chosen keywords and terms used in "Methods" for the idiopathic male fertility. In addition, the count of publications do not correspond to reported numbers and differ from the diagram (Fig. 1). I do not consider it necessary to analyze and present such information at all in review.

Answer to comment 2: Figure 1 was replaced and the number of publications was not mentioned. The search strategy was changed and only articles on seminal plasma proteome and transcriptome in male idiopathic infertility were included and discussed.

Comment 3: The tables show only the list of genes and proteins without context relationship to pathologies. The authors should describe everything in the text, evaluate and discuss. The individual subchapters contain very little information related to the topic and a small set of works is often cited.

Answer to comment 3: We have now provided more explanation for each study included in the present review.

Comment 4: Discussion is not absolutely sufficient and very short. Additionally, obtained information from cited papers are not properly concluded.

Answer to comment 4: No discussion paragraph is now mentioned since papers are discussed when they are mentioned. The last paragraph now includes a practical proposal of molecular targets more frequently reported (e.g. miR-34, miR-122, miR-509) or having an already shown predictive role (e.g. ECM1, TEX101, and LGALS3BP) to be validated by further studies.

Round 2

Reviewer 1 Report

No additional comments

Author Response

Thank you for the time spent to review this manuscript. We think that your comments have significantly improved its quality. 

Reviewer 2 Report

THe authors have addressed my concerns adequately

Author Response

We are grateful for the time spent to review this manuscript. 

Reviewer 3 Report

The authors have adjusted their manuscript to my comment satisfactorily. They focused only in general on the differences in transcriptome/proteome of normal and pathological SP. Now, the review is better arranged. I have minor comments:  Please define miR-34, 122, 509 as micro RNA in the Abstract.
I suggest to replace the title of subchapters "Evidence in male infertility" to Overview or Implication in ....

Author Response

Done as requested. Thank you.